# LC-MS and Transcriptome Analysis of Lipopeptide Biosynthesis by *Bacillus velezensis* CMT-6 Responding to Dissolved Oxygen

**DOI:** 10.3390/molecules27206822

**Published:** 2022-10-12

**Authors:** Qi Deng, Haisheng Lin, Meifang Hua, Lijun Sun, Yuehua Pu, Jianmeng Liao, Zhijia Fang, Saiyi Zhong, Ravi Gooneratne

**Affiliations:** 1Guangdong Provincial Key Laboratory of Aquatic Product Processing and Safety, Guangdong Provincial Engineering Technology Research Center of Seafood, Guangdong Province Engineering Laboratory for Marine Biological Products, Key Laboratory of Advanced Processing of Aquatic Product of Guangdong Higher Education Institution, College of Food Science and Technology, Guangdong Ocean University, Zhanjiang 524088, China; 2Collaborative Innovation Center of Seafood Deep Processing, Dalian Polytechnic University, Dalian 116034, China; 3Zhanjiang Institute of Food and Drug Control, Zhanjiang 524022, China; 4Department of Wine, Food and Molecular Biosciences, Faculty of Agriculture and Life Sciences, Lincoln University, P.O. Box 85084, Lincoln 7647, New Zealand

**Keywords:** surfactin and iturin, dissolved oxygen, transcriptomic, LC-MS

## Abstract

Dissolved oxygen (DO) is an key factor for lipopeptide fermentation. To better understand the link between oxygen supply and lipopeptide productivity in *Bacillus velezensis* CMT-6, the mechanism of DO on the synthesis of antimicrobial lipopeptides by *Bacillus velezensis* CMT-6 was examined. The production of surfactin and iturin of CMT-6 was detected by liquid chromatography–mass spectrometer (LC-MS) under different DO conditions and transcriptome analysis was performed. At 100 and 200 rpm, the lipopeptides productions were 2753.62 mg/L and 3452.90 mg/L, respectively. There was no significant change in the yield of iturin but that of surfactin increased by 64.14%. Transcriptome analysis revealed that the enriched differential genes were concentrated in the GO term of oxidation–reduction process. The marked enrichment of the lipopeptides synthesis pathway, including microbial metabolism in diverse environments and carbon metabolism in the two-component system, were observed. More importantly, the expression levels of the four surfactin synthetase genes increased at higher DO, however, the iturin synthetase gene expression did not. Furthermore, modular surfactin synthetase was overexpressed (between 9- and 49-fold) at 200 rpm but not at 100 rpm, which is suggestive of efficient surfactin assembly resulting in surfactin overproduction. This study provides a theoretical basis for constructing engineering strains with high lipopeptide production to adapt to different DO.

## 1. Introduction

Surfactin and iturin are the two most widely studied lipopeptides [1]. The molecular structure of the two cyclic lipopeptides mostly comprise seven amino acids and fatty acid chains (the fatty acid chain length ranges from C_13_ to C_16_ in surfactin and from C_14_ to C_17_ in iturin) [2]. Based on their unique structures, surfactin and iturin possess broad antimicrobial spectra because they can insert into a phospholipid bilayer to cause cell membrane perforation and destroy the integrity of cell wall and mycelium [2,3]. In addition, the lipopeptides are provided with strong surfactivity, excellent antibacterial activity and good foamability and emulsifiability [3,4]. Thus, they play an important role in many fields including food, medicine and cosmetic manufacture [5,6,7].

At present, lipopeptides are mainly produced by the *Bacilli* species. During fermentation, dissolved oxygen (DO) is an important process parameter because it affects the biomass, cell differentiation and electron transfer [8]. In addition, DO affects metabolic pathways and fluxes in *Bacilli* and hence lipopeptide production [9]. Lipopeptide production can be enhanced by increasing the DO availability at higher rotational speeds [10]. However, severe stirring causes serious foaming which leads to a loss of lipopeptides [11]. Therefore, it is of great significance to explore how DO regulates lipopeptide synthesis.

Both surfactin and iturin are synthesized via a systematic mechanism catalyzed by nonribosomal peptide synthetases (NRPSs) [12]. Surfactin is encoded by the *srfA* operon (*srfAA*, *srfAB*, *srfAC*, *srfAD*), and iturin is encoded by the itu operon (*itu A*, *ituB*, *ituC*, *ituD*) [2,13]. The synthesis of surfactin and iturin is not only regulated by synthetase genes but also mostly by the tricarboxylic acid cycle (TCA) pathway, fatty acid synthesis, amino acid synthesis and protein efflux [14]. However, the expression of these genes and pathways at different DO environments and their effects on lipopeptide production are still unknown.

In this study, LC-MS/MS was used to detect lipopeptides including surfactin and iturin at high and low DO produced at rotational speeds of 200 and 100 rpm, respectively. Comparative transcription was used to explore the metabolic pathways that respond to DO and regulate lipopeptide synthesis. This study provides new insights to better elucidate the signaling network between DO level and lipopeptide yields, and provide guidance to further improve lipopeptide production.

## 2. Results and Discussion

### 2.1. Phenotypic Assays for Biomass and Bacillus velezensis CMT-6 Lipopeptide Production

*Bacilli* produce lipopeptides by aerobic fermentation. The DO concentration in the fermentation broth not only affects bacterial growth but also changes the metabolic flow of lipopeptide synthesis [8,9]. The lipopeptide yield improves at higher DO concentrations. No significant change occurred in the iturin concentration (*p* > 0.05) but surfactin reached a maximum of 1702 mg/L at the higher DO concentration (Figure 1B). However, the biomass of CMT-6 did not increase at a higher DO. In fact, the dry cell weight of CMT-6 at a rotational speed of 200 rpm was significantly lower than that at 100 rpm (*p* < 0.05) (Figure 1A). It has been reported that when cells are grown vigorously, the lipopeptide production is lower [15]. Therefore, the fermentation process of increasing biomass is not an effective strategy to improve the production of lipopeptides in actual production.

### 2.2. Global Transcriptome Analysis

The transcriptome analysis based on the RNA sequencing of CMT-6 at the two DO culture concentrations showed the number of differentially expressed genes (DEGs) to be 795, among which 444 were up-regulated and 351 were down-regulated in the 200 rpm group compared with the 100 rpm group, while 3201 genes underwent no significant changes in expression (Figure 2).

According to the Gene Oncology (GO) analyses (Figure 3), the up-regulated DEGs were significantly enriched in the biological process, including transmembrane transport (40 genes: *cydD*, *ydjE*, *nhaC*, *pstC*, *mntH*, *NRT*, *narK*, *nrtP*, *nasA*, *araE*, *iolT*, *HXT*, *pbuX*, *gltP*, *gltT*, *msmX*, *msmK*, *malK*, *sugC*, *ggtA*, *msiK*, *dctA*, *mscS*, *fsr*, *iolF*, *ynfM*, *glcP*, *bcr*, *citM*, *glpT*, *lrgA*, *dgoT*, *bcr*, *ynaI*, *mscMJ*, *MSL*, *cydC*, *oppA*, *mppA*, *HXT*), ‘de novo’ UMP biosynthetic process (7 genes: *pyrB*, *pyrDI*, *pyrDII*, *pyrF*, *pyrE*, *pyrC*, *carA*), biotin biosynthetic process (5 genes: *bioB*, *bioW*, *bioA*, *bioK*, *bioD*), phosphoenolpyruvate-dependent sugar phosphotransferase system (13 genes: *celB*, *ptsI*, *celA*, *mtlA*, *celA*, *lacF*, *celB*, *celC*, *gamP*, *glvB*, *FruB*, *treB*, *celC*), histidine biosynthetic process (10 genes: *hisIE*, *hisH*, *E3.1.3.15B*, *hisF*, *hisB*, *hisZ*, *hisD*, *hisG*, *hisA*, *hisC*), oxidation–reduction process (103 genes: *gldA*, *queF*, *ldh*, *glpA*, *glpD*, *pyrDI*, *pdhA*, *leuB*, *qcrC*, *bfcC*, *petD*, *mmsA*, *iolA*, *narH*, *celF*, *iolX*, *aroE*, *glcD*, *dps*, *pdhB*, *kefG*, *paaH*, *hbd*, *fadB*, *mmgB*, *pyrDII*, *betB*, *gbsA*, *araM*, *egsA*, *butB*, *bkdA2*, *iolG*, *adhP*, *katE*, *catB*, *srpA*, *rspB*, *nirB*, *zwf*, *hemY*, *lpd*, *pdhD*, *gapA*, *frmA*, *adhC*, *cydA*, *tyrA2*, *wbpA*, *melA*, *ilvC*, *gnd*, *gntZ*, *serA*, *bacC*, *fldA*, *nifF*, *isiB*, *l pd*, *pdhD*, *tdh*, *bdh*, *lpd*, *pdhD*, *bacG*, *dps*, *narI*, *ttuC*, *dmlA*, *coxD*, *ctaF*, *efeB*, *proA*, *hisD*, *glvA*, *gudB*, *rocG*, *osmC*, *dadA*, *bkdA1*, *cobA*, *hemD*, *mtlD*, *fldA*, *nifF*, *isiB*, *gbsB*, *gabD*, *serA*, *katE*, *catB*, *srpA*, *glcF*, *acuA*, *gcvPB*, *ME2*, *sfcA*, *maeA*, *bioI*, *nirD*, *gapA*, *ald*, *ssuD*), carbohydrate transmembrane transport (12 genes: *CelB*, *celA*, *mtlA*, *celA*, *celC*, *celB*, *lacF*, *glvB*, *mtlA*, *fruB*, *treB*, *celC*). These results show that the effects of DO on the synthesis of lipopeptides by CMT-6 involve a range of physiological functions of CMT-6. The enriched differential genes were mostly concentrated in the transmembrane transport and oxidation–reduction processes. More genes with transmembrane transport function means that more protein translocation channels could be synthesized, including those involved in the secretion of lipopeptides, extracellularly resulting in an increase in production [16,17]. A greater number of genes with oxidation–reduction processes reduces the conversion of NADH into NAD to ensure the oxidation–reduction state (NADH/NAD^+^) balance at high DO concentrations [18], resulting in more ATP production with an increased utilization of substrate by bacteria, culminating in an accelerated metabolism to the synthesis of more lipopeptide [9,19].

Genes usually play a role in specific biological functions by interacting with each other. Pathway analysis can provide a deeper understanding of the biological functions of CMT-6 genes. A scatter plot of the KEGG enrichment analysis of CMT-6 at two DO concentrations (in which only the top 20 enriched pathway entries are shown) is shown in Figure 4. The differential genes of the two DO groups were significantly enriched in three pathways—microbial metabolism in diverse environments, carbon metabolism and two-component system. These metabolic pathways are closely related to lipopeptide synthesis. Firstly, the high gene expressions of those enriched in the metabolism including the carbon metabolism pathway probably acted to reduce the delay period for optimal nutrient utilization in the fermentation system and to provide sufficient precursors and energy for lipopeptide synthesis [20]. Secondly, the two-component system (TCS) can respond to external environment changes [21] and many genes could be regulated by TCS with their functions involving phosphatase synthesis, anionic polymer formation, phosphoteic acid and the synthesis of secondary metabolites [22,23,24]. There are other genes that can combine with TCS, including lipopeptide synthetase genes [25,26,27]. The expression levels of spore and biofilm regulatory genes enriched in TCS were up-regulated, which provides the most direct theoretical support for the development of high-yield fermentation processes such as the biofilm method under hypoxic conditions.

### 2.3. The Effect of Dissolved Oxygen on Synthetase Genes and the Key Genes Associated with Lipopeptides

The synthesis of surfactin and iturin were accomplished by non-ribosomal peptide synthetase, which are encoded by *srfA* and *itu*, respectively [14,28,29]. In this study, *srf*AA-*srfAD* were up-regulated at a high DO (Figure 5), which improved the surfactin yield. The expression levels of *ituA*-*ituD* were relatively stable (Figure 5), which resulted in no significant change in the iturin yield. Lipopeptide production is reduced when the expression of synthetase genes is inhibited [30]. However, there is also a different view that the expression of synthetase genes of the strains with high-yield lipopeptide is lower than that in the low-yield strains [14]. Therefore, there is no consistent conclusion about the relationship between the synthetase gene expression levels and the ability to synthesize lipopeptides. A possible reason is that synthetases contain multiple enzyme subunits, each containing multiple functional modules which are responsible for the activation of specific amino acids and the extension of peptide chains [30]. Therefore, the ability to synthesize lipopeptides could be determined by the synthetase gene content and its catalytic efficiency to join amino acids.

There are 14 TCA genes that respond to the changes in DO concentrations (Table 1). The gene expressions in the 200 rpm group were higher than in the 100 rpm group. The increase in DO enhanced the metabolic flow of TCA to synthesize more amino acids and fatty acids, which are the precursors for lipopeptide synthesis [31,32,33,34]. Based on these findings, the engineering strains could be constructed with the high expression level of these genes to improve the utilization efficiency of carbon sources.

Fatty acid is the main component of lipopeptide and the gene expression level in its synthesis path is positively correlated with lipopeptide production [35,36]. Certain associated genes such as *accC*, *accB*, *acsL and fadD* were also up-regulated at high DO (Table 2). This is probably the reason that CMT-6 can synthesize high levels of lipopeptide at high DO.

The elevated nitrogen metabolism gene expression is for the bacillus to adapt to survive under hypoxic conditions [37]. The expressions of *narG*, *narH*, *narI*, *narJ*, *liaF*, *liaG*, *liaH* and *liaI* genes were down-regulated at high DO concentrations (Table 3). This gives us inspiration: in order to reduce the dependence of the fermentation process of lipopeptides at a high DO, the expression of genes related to nitrogen metabolism could be promoted by genetic engineering technology, so the production of lipopeptides would be enhanced.

## 3. Materials and Methods

### 3.1. Experimental Strain and Medium

*Bacillus velezensis* CMT-6 (Gen Bank, CP025341) culture was obtained from the Food-borne Pathogenic Microorganisms and Toxins of Aquatic Products Green Control Laboratory, Food Science and Technology College, Guangdong Ocean University, Zhanjiang, China.

### 3.2. Cell Growth and Lipopeptide Production Assay

CMT-6 was inoculated to 100 mL LB liquid medium, cultured at 37 °C and centrifuged at 150 rpm for 24 h for seed preparation. The seed solution was added to the modified Landy (1.0 g/L K_4_H_2_PO_4_, 0.5 g/L KCl, 0.5 g/L MgSO_4_·7H_2_O, 20 g/L glucose, 1 g/L yeast extract, 0.0016 g/L CuSO_4_·5H_2_O, 0.0015 g/L FeSO_4_·7H_2_O, 0.05g/L MnSO_4_, 5 g/L L-Glutamic acid (L-Glu)) medium at a concentration of 5 % (*v*/*v*) in an Erlenmeyer flask and cultivated at 30 °C—one half at 100 rpm and the remainder at 200 rpm for 36 h. After cultivation, each CMT-6 culture was centrifuged at 5000 rpm (J2-MC, Beckman, Brea, CA, USA) for 15 min at 4 °C. The thallus was collected, freeze-dried and weighed. The supernatant was stored at 4 °C.

The crude extracts were mixed with isovolumetric acetonitrile/water (7:3, *v*/*v*) in 0.1 % (*v*/*v*) formic acid, and filtered with a 0.45 μm biofilter into the autosampler vials for the measurement of surfactin and iturin as follows [38].

Surfactin and iturin analyses were performed on a Thermo Scientific Surveyor HPLC system comprised of a Surveyor MS Pump Plus, an on-line degasser and a Surveyor auto sampler Plus coupled with a Thermo TSQ Quantum Access tandem mass spectrometer equipped with an electrospray ionization (ESI) source (Woburn, MA, USA). The separation was performed at 35 °C using a Hypersil GAcquity UPLC@BEM C18 column (5 μm, 250 mm × 4.6 mm) (Thermo Scientific, Carlsbad, CA, USA) with a flow rate of 5.0 μL/min. The mobile phase consisted of acetonitrile (A) and water containing 5 mM ammonium acetate 0.1 % formic acid (B) with the gradient elution program as follows: 0–0.3 min 45% A, 0.3–0.6 min 50% A, 0.6–1.8 min 80% A and 6 min 100% A. MS/MS detection was carried out using a triple quadruple mass spectrometer, coupled with an electrospray ionization source operated in positive (ESI+) mode (Shimadzu, Kyoto, Japan). The ionization source parameters were set as follows: capillary voltage—1.2 KV; ion source temperature—150 °C; spray temperature—450 °C; desolvent gas flow rate—600 L HR-1; impact energy—6.0 eV; molecular weight deviation within ±0.2 Da; and mass charge ratio range from 800 to 2000.

### 3.3. RNA Extraction and Analysis of RNA Sequencing Data

#### 3.3.1. RNA Extraction

The thallus was obtained according to the method of 2.2, and immediately transferred to liquid nitrogen and stored at −80°C for RNA extraction. Total RNA was extracted using Trizol reagent (Invitrogen Life Technologies, Carlsbad, CA, USA) according to the manufacturer’s instructions. The integrity of the extracted RNA sample was determined by electrophoresis. The RNA concentration was measured using a NanoDrop spectrophotometer (Thermo Scientific, Waltham, MA, USA).

#### 3.3.2. Construction of the cDNA Library and Transcriptomic Data Analysis

Sequencing libraries were generated using the TruSeq RNA Sample Preparation Kit (Illumina, San Diego, CA, USA) according to the manufacturer’s instructions. Briefly, mRNA was purified using poly-T oligo attached magnetic beads. The mRNA was fragmented using divalent cations at high temperature in an Illumina proprietary fragmentation buffer. First-strand cDNA was synthesized using random oligonucleotides and SuperScript II. Second-strand cDNA was subsequently synthesized using DNA polymerase I and RNase H. After adenylation of the 3′ ends of the DNA fragments, Illumina PE adapter oligonucleotides were ligated, and the library fragments purified were using the AMPure XP system (Beckman Coulter, Brea, CA, USA). DNA fragments with ligated adaptors on both ends were selectively enriched using Illumina polymerase chain reaction (PCR) Primer Cocktail in a 15 cycle PCR reaction. The products were purified (AMPure XP system) and quantified using the Agilent high-sensitivity DNA assay with the Bioanalyzer 2100 system (Agilent Technologies, Santa Clara, CA, USA). The sequencing libraries were sequenced using Illumina HiSeq.

### 3.4. Data Analysis

The transcriptome library of each sample obtained by high-throughput sequencing was converted into original sequence data, and the CASAVA base sequence identification analysis was performed. The valid data in this study were obtained by filtering adapters, aggregating N and low-quality readings from the original data. Then, the software HISAT2 V2.0.5 was used to compare the filtered RNA sequence with the reference genome of Bacillus Velez downloaded from NCBI (https://www.ncbi.nlm.nih.gov/g accessed on 25 August 2022). The HTseq software was used to process the data generated by high-throughput sequencing, and these reads were efficiently and accurately compared to the genes to estimate the gene expression levels of different genes and different comparable experiments. DESeq 2R software was used to analyze and compare the differentially expressed genes. The Benjamini and Hochberg method was used to adjust the *p* value, and *p* < 0.05 and log2 fold change2 were set together to analyze differentially expressed genes (DEGs).

### 3.5. Enrichment Analysis

In order to analyze the function of differentially expressed genes, genes with a correction value of *p* ≤ 0.05 were used as significantly rich GO terms, and all differentially expressed genes were mapped to Gene Ontology terms in the database. In order to determine the key metabolic pathways, the cluster Profiler R software was used to analyze the differentially expressed genes in the KEGG pathway.

### 3.6. Statistical Analysis

The data are expressed as the mean ± standard deviation (SD) and were statistically analyzed by IBM SPSS statistics 26.0 software. Significant differences between the control and the treated fish were determined by one-way analysis of variance (ANOVA), followed by Tukey’s test to compare the control and treatment group values. A *p*-value of <0.05 was considered significant. The figures were constructed by origin 9.0.

## 4. Conclusions

Transcriptomics and LC-MS were used to reveal the molecular mechanism of lipopeptide accumulation by CMT-6 responding to DO concentration. The high dissolved oxygen promoted metabolism flows more towards the synthesis of precursors, and the high expression of lipopeptide synthase genes increased the utilization of precursor substances by the strain, thus increasing the production of lipopeptides. These findings provide theoretical guidance to construct engineered bacterial strains and the development of hypoxic fermentation strategies to increase the lipopeptides yield. However, the differential responses of different lipopeptides’ synthetase genes to DO need to be further studied.

## Figures and Tables

**Figure 1 molecules-27-06822-f001:**
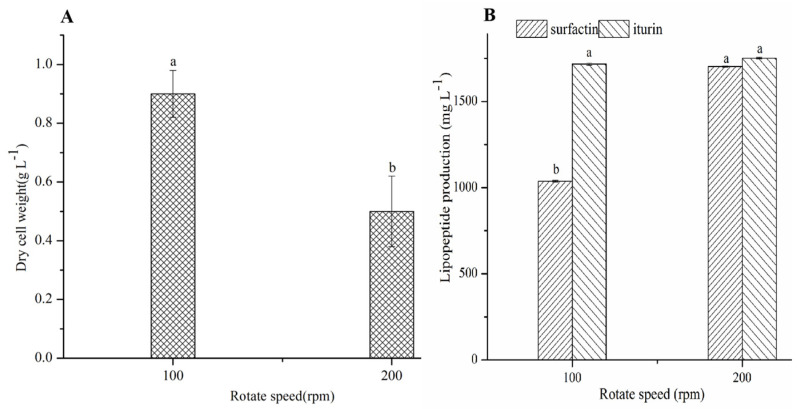
Biomass (**A**) and lipopeptide production (**B**) of *B. velezensis* CMT-6. a, b represents the difference between different treatment groups. The same letter represents no significant difference, while different letters represent significant difference.

**Figure 2 molecules-27-06822-f002:**
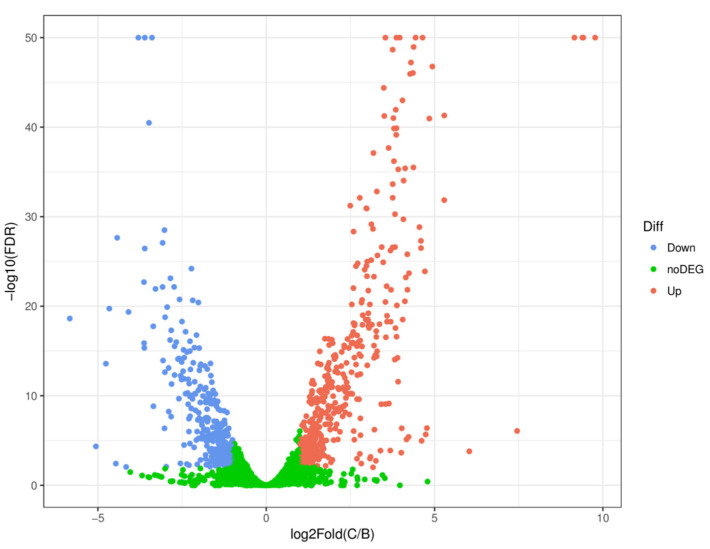
Volcano plot of differentially expressed genes in *Bacillus velezensis* CMT-6 at high and low dissolved oxygen concentrations. Each dot represents a gene: red dots represent up-regulated genes; blue dots represent down-regulated genes; and green dots represent differentially expressed genes for which no significant change was observed.

**Figure 3 molecules-27-06822-f003:**
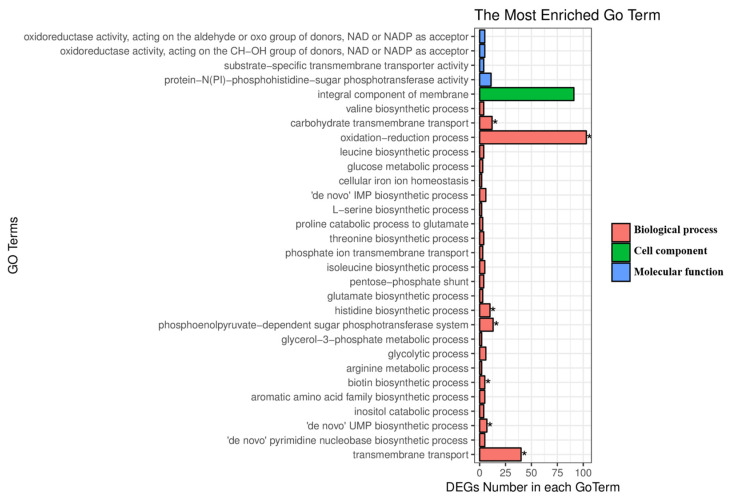
The GO classification of the differentially expressed genes by *Bacillus velezensis* CMT-6 at the higher DO concentration (* represents genes with significantly enriched GO terms (*p*-value ≤ 0.05).

**Figure 4 molecules-27-06822-f004:**
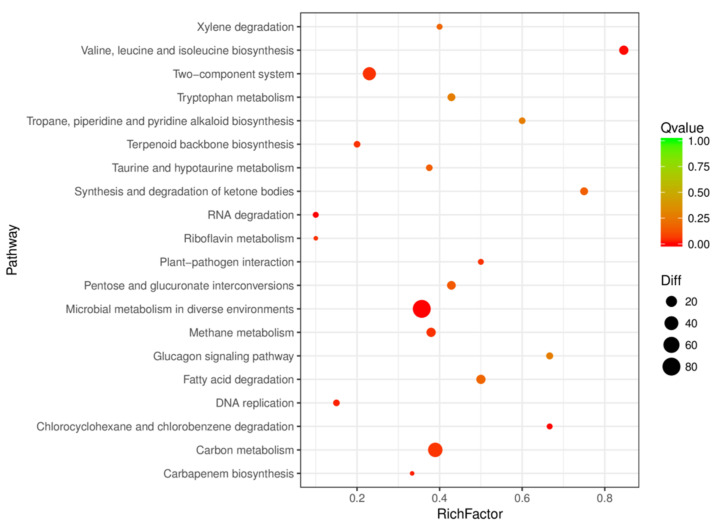
KEGG classification of the differentially expressed genes of *Bacillus velezensis* CMT-6. RichFactor refers to the ratio of the number of differentially expressed genes annotated in the pathway term.

**Figure 5 molecules-27-06822-f005:**
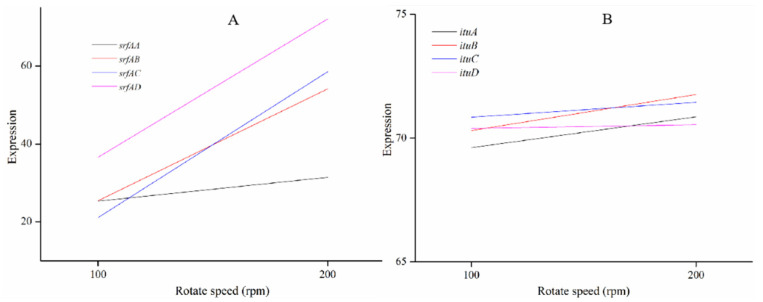
The expression of lipopeptides, surfactin (**A**) and iturin (**B**) biosynthesis genes in *Bacillus velezensis* CMT-6.

**Table 1 molecules-27-06822-t001:** The TCA gene expression in *Bacillus velezensis* CMT-6 at different dissolved oxygen concentrations.

Gene	Gene Expression		Gene Function
	100 rpm	200 rpm	
*pckA*	43.66	129.17	phosphoenolpyruvate carboxykinase (ATP)
*aceF*, *pdhC*	6.10	3284.60	dihydrolipoamide acetyltransferase
*pdhA*	5.75	2596.92	pyruvate dehydrogenase E1 component beta subunit
*pdhB*	2.74	1931.38	
*lpd*	8.72	4663.19	dihydrolipoamide dehydrogenase
*pdhD*	39.09	82.20	
*mdh*	725.33	1505.83	malate dehydrogenase
*gltA*	511.65	1315.62	citrate synthetase
*acnA*	349.26	811.09	aconitate hydratase
*sdhB*, *frdB*	399.27	582.47	succinate dehydrogenase
*sucB*	438.12	783.63	dihydrolipoamide succinyltransferase
*icD*	683.05	1187.85	isocitrate dehydrogenase

**Table 2 molecules-27-06822-t002:** Gene expression of fatty acid biosynthesis in *Bacillus velezensis* CMT-6 at different dissolved oxygen concentrations.

Gene	Gene Expression		Gene Function
	100 rpm	200 rpm	
*accC*	52.23	169.80	acetyl-CoA carboxylase, biotin carboxylase subunit
*accB*	4.77	14.25	acetyl-CoA carboxylase biotin carboxyl carrier protein
*acsL*	77.05	151.61	long-chain acyl-CoA synthetase
*fadD*	83.39	377.19	long-chain acyl-CoA synthetase

**Table 3 molecules-27-06822-t003:** The gene expression of nitrogen metabolism in *Bacillus velezensis* CMT-6 at different dissolved oxygen concentrations.

Gene	Gene Expression		Gene Function
	100 rpm	200 rpm	
*kinC*	79.87	47.95	sporulation sensor kinase C
*kinD*	90.39	33.96	sporulation sensor kinase D
*narG*	19.12	1.83	nitrate reductase alpha subunit
*narH*	7.29	1.27	nitrate reductase beta subunit
*narI*	12.36	3.01	nitrate reductase gamma subunit
*narJ*	8.20	2.72	nitrate reductase delta subunit

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
