# Peer review of "LC-MS and Transcriptome Analysis of Lipopeptide Biosynthesis by Bacillus velezensis CMT-6 Responding to Dissolved Oxygen"

_molecules, 2022, doi:10.3390/molecules27206822_

Round 1

Reviewer 1 Report

In this study the authors used LC-MS/MS monitor the production of surfactin and iturin related to dissolved oxygen (low vs. high) at two rotational speeds. The corresponding pathways and mechanisms were also explored and discussed. The presentation was neat and organized. Easy to follow.

Author Response

Thank you very much for your letter with comments to improve the manuscript quality. We also would like to express our sincere thanks to you for the constructive and positive comments.

Reviewer 2 Report

Comments to the authors  for evaluating the following manuscript

Title:   

 Regulation of Oxygen Availability on Lipopeptide 2 Biosynthesis by Bacillus velezensis CMT-6 based on 3 LC-MS and Transcriptomics 

Title: the title should be rephrased to reflect the outcome of this work

Abstract section

·         The structure of the abstract was very poor, I known the restriction in the number of words (200); meanwhile we can provide the important results only. Please start the abstract with sentences which reflect the necessary of this work

·         In line 21, Please provide the abbreviation of  LC-MS

·         Please add the overall practical implementation of your results or other hypothesis that may be utilized in the future at the end of the abstract

Introduction section

·         Please in concise manner add a paragraph about the antimicrobial activities of lipopeptides and focus on  Surfactin and iturin  

·         Please if you can to elaborate on the introduction, the necessary of the increasing the production of lipoproteins commercially  

·         The objective of the study at the end of introduction wasn't clear "this is the title of the manuscript"

Methodology

·         Please provide all programs which used to construct all figures in this manuscript

Results

The data in tables 1,2 and 3 were repeated in the figure 5 therefore, these table must be deleted and the genes function must be added in the introduction section (this is not your results)

The quality of figures 1,2,3 and 4 were very poor and I cannot read them

Discussions

·         Although the discussion was merged with the results; meanwhile, it need numerous modification. It should focus on explaining and evaluating what you found (the main results), showing how it relates to the new researches

Conclusion section:

·         It must be rephrased, conclusion section must provide us with the applied implication of your results in concise manner

·         Please add at the end of the manuscript the limitations: what can’t the results and discussion tell us?

Reviewer 3 Report

 The manuscript ( Regulation of Oxygen Availability on Lipopeptide Biosynthesis by Bacillus velezensis CMT-6 based on LC-MS and Transcriptomic) is interesting but I believe that it should be improved based on the following points.

I think it was necessary to check the potential of B. velezensis as biological control agents under higher dissolved oxygen levels.

I think also it is important to discuss the roles of other genes from your transcriptomic analysis results.

The introduction and discussion of the results are very short!

L31 Please make B. velezensis italic

L48 Please correct to However, severe stirring causes serious foaming leading to a loss of lipopeptides.

L50 Please correct the spelling (a complex mechanism)

L71 Please correct to at a rotational speed

In your opinion what are the applications of your study results?

Round 2

Reviewer 2 Report

thanks my dear authors, all my recommendation were well done and additional information were inserted. therefore, i have no  other criticisms or recommendation 

Reviewer 3 Report

Thank you for your answers.